

# DPIF-Net: a dual path network for rural road extraction based on the fusion of global and local information

Yuan Sun[1], Xingfa Gu[1], Xiang Zhou[1], Jian Yang[1], Wangyao Shen[2], Yuanlei Cheng[2], Jin Ming Zhang[2] and Yunping Chen[2]

[1] Aerospace Information Research Institute, Chinese Academy of Sciences, Beijing, China
[2] School of Automation Engineering, University of Electronic Science and Technology of China, Chengdu, Sichuan, China

## ABSTRACT

**Background**. Automatic extraction of roads from remote sensing images can facilitate many practical applications. However, thus far, thousands of kilometers or more of roads worldwide have not been recorded, especially low-grade roads in rural areas. Moreover, rural roads have different shapes and are influenced by complex environments and other interference factors, which has led to a scarcity of dedicated low level category road datasets.

**Methods**. To address these issues, based on convolutional neural networks (CNNs) and tranformers, this article proposes the Dual Path Information Fusion Network (DPIF-Net). In addition, given the severe lack of low-grade road datasets, we constructed the GaoFen-2 (GF-2) rural road dataset to address this challenge, which spans three regions in China and covers an area of over 2,300 km, almost entirely composed of low-grade roads. To comprehensively test the low-grade road extraction performance and generalization ability of the model, comparative experiments are carried out on the DeepGlobe, and Massachusetts regular road datasets.

**Results**. The results show that DPIF-Net achieves the highest IoU and $F_1$ score on three datasets compared with methods such as U-Net, SegNet, DeepLabv3+, and D-LinkNet, with notable performance on the GF-2 dataset, reaching 0.6104 and 0.7608, respectively. Furthermore, multiple validation experiments demonstrate that DPIF-Net effectively preserves improved connectivity in low-grade road extraction with a modest parameter count of 63.9 MB. The constructed low-grade road dataset and proposed methods will facilitate further research on rural roads, which holds promise for assisting governmental authorities in making informed decisions and strategies to enhance rural road infrastructure.

Corresponding authors
Xingfa Gu, guxf@aircas.ac.cn
Yunping Chen, chenyp@uestc.edu.cn

## INTRODUCTION

Roads are typical landscape features with complex topological relationships, and millions of kilometers of roads in the world are still unrecorded, particularly low-grade roads in rural areas. In China, low-grade roads are defined as those with an annual average daily

traffic volume of fewer than 6,000 cars (*China, 2003*). These roads are vital for promoting urban–rural economic exchange and narrowing the gap between urban and rural areas. Therefore, it is imperative to design an intelligent and automatic method for rural road extraction.

Although high-resolution remote sensing images have been studied for many years for road extraction, accurately identifying rural roads in these images may face additional challenges in reality (*Li et al., 2021*). High-resolution images provide rich discriminative features for road identification but also contain many interference factors. For instance, shadows on roads are produced due to occlusions from various vehicles, trees, and tall buildings under different illumination conditions. Rural roads are often characterized by an absence of distinct geometric features, and their connectivity can be impacted by nearby rivers, which in turn affects the effectiveness of road extraction. In addition, rural roads made of dirt are more difficult to extract than roads made of asphalt or cement.

To solve these problems, this article proposes an end-to-end network called the Dual Path Information Fusion Network (DPIF-Net), which combines the strengths of convolutional neural networks (CNNs) and transformers to further improve the accuracy of rural road extraction. Furthermore, since there are few datasets related to rural roads, a dataset of rural roads is specifically constructed. Finally, we present extensive experiments conducted on the DeepGlobe and Massachusetts datasets as well as our dataset to test the model's generalization ability and robustness.

The contributions of this article are summarized as follows:

(1) The proposed DPIF-Net, which has a small number of network parameters and a simple structure. It effectively combines the advantages of CNNs in spatial induction with the adaptive weighting of input information in transformers to establish global dependencies. Moreover, DPIF-Net can effectively extract both the local detailed features and global context features of roads and fully integrate this information to produce more accurate road segmentation results.

(2) The constructed dataset of rural roads. Our dataset includes roads of different regions in China, but most of them are various types of rural roads. This dataset is specifically constructed for studying rural road extraction and our model's performance on rural roads.

The rest of this article is organized as follows. Section 'Related Works' describes some related work on deep learning for road extraction. Section 'Materials & Methods' explains the road dataset used in the experiments and describes the details of the method proposed in this article. Section 'Results' presents the experimental results and analysis. Sections 'Discussion' and 'Conclusions' provide a discussion and conclusions.

## RELATED WORKS

At present, road extraction and monitoring operations are still performed manually or semimanually, making them ineffective and costly (*Abdollahi et al., 2020*). Therefore, new robust techniques, such as deep learning methods, are needed to accurately extract road networks of various scales from remote sensing imagery (*Panboonyuen et al., 2017*),

which has gradually become a prominent direction of research. With the development of artificial intelligence, deep convolutional neural networks (DCNNs) are gradually gaining dominance in the field of image processing. In recent years, there has been an explosion of various papers on road segmentation with DCNNs, and many excellent CNN models have emerged, such as U-Net (*Ronneberger, Fischer & Brox, 2015*), LinkNet (*Chaurasia & Culurciello, 2017*), SegNet (*Badrinarayanan, Kendall & Cipolla, 2017*), D-LinkNet (*Zhou, Zhang & Wu, 2018*), DeepLabv3+ (*Chen et al., 2018b*), and generative adversarial networks (GANs) (*Goodfellow et al., 2020*). These models integrate features from multiple layers of a CNN to exploit the multiscale information at different semantic levels (*Zhu et al., 2021*). Many road segmentation methods are based on the above models.

*Zhang, Liu & Wang (2018)* integrated residual units into a U-Net-like network for road extraction. Residual units can make it easier for a network to learn features and achieve better results. *Moradi et al. (2019)* proposed a modified U-Net architecture combined with a feature pyramid network and concatenated the feature maps from all levels of the U-Net decoder path as input. Their method achieved good performance in medical image segmentation. *Chen et al. (2021)* proposed a reconstruction bias U-Net for road extraction from remote sensing images. This method obtains multiple levels of semantic information from different upsampling scales by adding decoding branches. However, the extraction effect of the modified method is not good for low-grade roads, such as rural roads. *Yang et al. (2019)* constructed a U-Net network consisting entirely of Region CNN (RCNN) blocks, which preserve rich low-level spatial features. Inspired by U-Net and atrous spatial pyramid pooling (ASPP) *Chen et al., 2018a*, *He et al. (2019)* integrated an ASPP module into U-Net to obtain multiscale road information. *Lu et al. (2019)* proposed a deep learning framework based on U-Net, which can extract roads and road centerlines, and integrate feature information from different scales to improve the robustness of the model. *He et al. (2019)* added ASPP between the encoder and decoder in U-Net. At the same time, a loss function that considered the digital number (DN) value, contrast, structure and other factors of the image was proposed. *Lu et al. (2019)* replaced the first convolutional layer of each group in U-Net with a multiscale module and constructed a pyramid-like structure to complete the extraction of roads and road centerlines. To capture more information, a weighted loss for roads and centerlines was built. Each loss component was weighted in accordance with the relative proportions of background and target to solve the problem of target class imbalance.

Based on LinkNet, *Wang, Seo & Jeon (2021)* proposed an efficient nonlocal LinkNet with nonlocal blocks (NLBs) that can grasp relations between global features. This enables each spatial feature point to refer to all other contextual information and results in more accurate road segmentation. *Zhu et al. (2021)* added an attentive GCA block between the encoder and decoder to make the extracted road information more complete. They used FRN normalization to improve the robustness of the model. *Xie et al. (2019)* replaced the D-LinkNet intermediate structure with a global perception block for higher-order information. The design of the high-order information global perception block was inspired by bilinear pooling. Experiments showed that it achieved better performance than atrous convolution and could reduce the number of parameters by 1/4. *Zhu et al. (2020)*

proposed a model based on D-LinkNet and conditional random fields (CRFs) to solve the edge smoothing problem in the process of building extraction.

*Tao et al. (2019)* proposed a network composed of a spatial information inference structure (SIIS) for road extraction, and the overall framework was based on DeepLabv3+. The SIIS consisted of two groups of RCNN units. A weighted loss function combining the mean squared error (MSE) and intersection over union (IoU) was adopted. To solve the problem of imbalanced samples, images with excessively small target proportions relative to the background were removed. *Lourenco et al. (2023)* proposed combining DeepLabv3+ with an optimization strategy to extract rural roads.

Many road extraction methods based on generative adversarial network (GAN) have achieved impressive results. *Zhang et al. (2019b)* proposed a GAN for road extraction that had multiple discriminators. In the experiments, it was found that a combination of four discriminators and one generator was best. At the same time, a road label generation method that needed less manual intervention was proposed. *Shamsolmoali et al. (2021)* integrated feature pyramids into GANs for road detection. *Zhang et al. (2019a)* explored different types of GANs. An end-to-end model for road extraction based on GANs was proposed. The influence of convolution kernels of different sizes was discussed, and it was concluded that large convolution kernels were not needed to improve the receptive field for road extraction.

In addition to the above models, some scholars have used other road extraction methods and have also achieved promising results. *Bastani et al. (2018)* presented a method to extract road networks based on iterative graph construction. The final road map was generated by iteratively adding new candidate road regions. A decision function was used to determine whether a candidate area is a road by training a CNN. However, this method requires knowledge of the initial points of the roads. *Shao et al. (2021)* proposed a two-task end-to-end CNN named the Multitask Road-related Extraction Network (MRENet) for road surface extraction and road centerline extraction. The network design of MRENet uses atrous convolutions and a pyramid scene parsing pooling (PSP pooling) module to expand the network's receptive field, integrate multilevel features, and obtain more abundant information. In addition, the authors used a weighted binary cross-entropy function to alleviate the background imbalance problem. *Zhang & Wang (2019)* introduced a network consisting of dense cavity convolution modules for road and building extraction. *Batra et al. (2019)* proposed joint learning based on orientation and segmentation maps to enhance the connectivity rate in road extraction. The CNN-based structure achieved good road extraction results, but the accuracy was not high for complex road networks, and the method was not effective for low-grade roads.

The transformer model has made a vital difference in the natural language processing (NLP) field because of its attention mechanism (*Vaswani et al., 2017*). Inspired by the powerful representation capabilities of transformers, researchers have extended transformers to computer vision tasks (*Han et al., 2020*). Compared with other networks, transformer-based networks can achieve comparable performance with less computation. *Dosovitskiy et al. (2020)* built a framework consisting of a pure transformer for image classification tasks. The architecture was trained using large-scale data to obtain pretrained

models. When transferred to vision tasks, it achieved a performance comparable to that of CNNs. *Xie et al. (2021)* combined a fully convolutional network with an attention mechanism to learn information from long-range contexts and achieved good results in image segmentation tasks. In addition, *Xie et al. (2021)* built a semantic segmentation framework combining transformer and multi-layer perceptron (MLP). The framework was simple, efficient and powerful and consisted of a hierarchical transformer encoder and a decoder composed of MLPs. It could output multiscale features and did not require positional encoding, resulting in significantly improved performance and efficiency compared with similar algorithms. Therefore, a structure based on the transformer demonstrates a clear advantage in global feature extraction. In summary, current research is predominantly based on CNN, which has shown good performance in typical road. However, the accuracy of this approach tends to diminish in complex road networks. Given the complexity of low-level road details, this article aims to explore the potential of combining CNN and transformer architectures specifically for low-level road contexts.

## MATERIALS & METHODS

### Dataset

Although the currently available public road datasets cover a wide range of road categoriesin cities, suburbs and rural areas in many countries worldwide, they contain many normal roads and few rural roads. Therefore, they are not suitable for analysis with a special focus on rural roads, but they can be used as test data for model generalization performance.

In this study, a rural road dataset was constructed based on the GaoFen-2 (GF-2) satellite. The GF-2 satellite carries a range of sensors, including a Panchromatic and Multispectral sensor (PMS), a wide-field-of-view sensor (WFV), and a hyperspectral sensor (HSI), which provide high-resolution imagery with spatial resolutions ranging from 0.8 m to 16 m. The images for the PMS sensor (450 to 900 nm) at 0.8 m were utilized in this article. The images include three regions covering an area of over 2,300 square kilometers: the junction between Jiancaoping District and Gujiao City in Shanxi characterized by imagery measuring $36,500 \times 34,258$ pixels, covering an approximate area of 752 square kilometers, the junction between Anyang and Shijiazhuang cities in Hebei characterized by imagery measuring $39,695 \times 31,311$ pixels, covering an approximate area of 795 square kilometers, and the junction area of Guangzhou and Foshan in Guangdong characterized by imagery measuring $36,020 \times 32,431$ pixels, covering an approximate area of 747 square kilometers. For example, the study area at the junction between Taiyuan and Jiancaoping District and mask samples, as illustrated in Fig. 1. All GF-2 data in our study is sourced from the China Centre for Resources Satellite Data and Application.

Nevertheless, the considerable size of the satellite images poses challenges in terms of efficient data loading, potentially leading to a substantial increase in training duration. Moreover, the absence of masks within the original dataset introduces complexities in conducting effective supervised training. To overcome these challenges, the images underwent cropping to achieve dimensions of $512 \times 512$ pixels initially. Subsequently, manual annotation based on image texture was executed to generate corresponding masks.

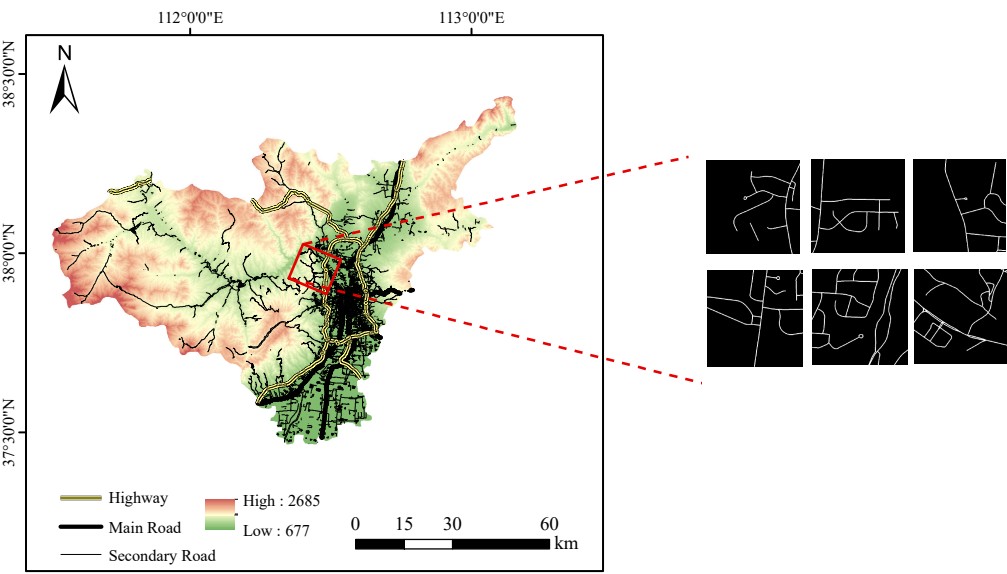

**Figure 1** **Partial study area schematic diagram and masks.** The ASTER GDEM data is open source and available at https://search.earthdata.nasa.gov.

In the end, we generated a dataset similar to the examples shown in Fig. 2. To address the potential sample imbalance, our dataset excluded the images containing abundant higher levelroads, consequently encompassing intricate details of rural road attributes, such as tree coverage, agricultural field irrigation channels, and road incompleteness. These adjustments were intended to enhance the model's performance in extracting low-grade rural roads. After preprocessing the data, 5,501 samples remained, with 5,421 as training samples, 40 as validation samples, and 40 as test samples.

In addition, experiments were carried out on two public datasets, DeepGlobe (*Demir et al., 2018*) and Massachusetts (*Mnih, 2013*). However, it is imperative to emphasize that these two datasets contain a limited quantity of low-grade roads in comparison to a substantial volume of regular highways. They are specifically employed to enhance our model's extraction performance and validate findings. The images in the DeepGlobe road dataset come from three countries, namely, India, Thailand, and Indonesia, and include multiple imaged scenes covering an area of over 2,220 square kilometers, such as cities, villages, wilderness, suburbs, seashores, and tropical rainforests. The ground resolution of the images is 0.5 m per pixel, and the image size is 1,024 × 1,024 pixels. There are a total of 6,226 images, of which 4,976 are designated for training and 1,250 are designated for testing. In this study, following *Zhu et al. (2021)*, the original images were cropped to a resolution of 512 × 512 pixels with an overlap of 256 pixels. Finally, a total of 5,000 images for training, 40 images for validation and 4,500 images for testing were obtained.

The Massachusetts road dataset consists of 1,171 aerial images of the Massachusetts region, which cover a wide variety of urban, suburban, and rural regions and an area of over 2,600 square kilometers. With a spatial resolution of 1 m per pixel, the images in this dataset have a size of 1,500 × 1,500 pixels and are composed of red, green, and blue

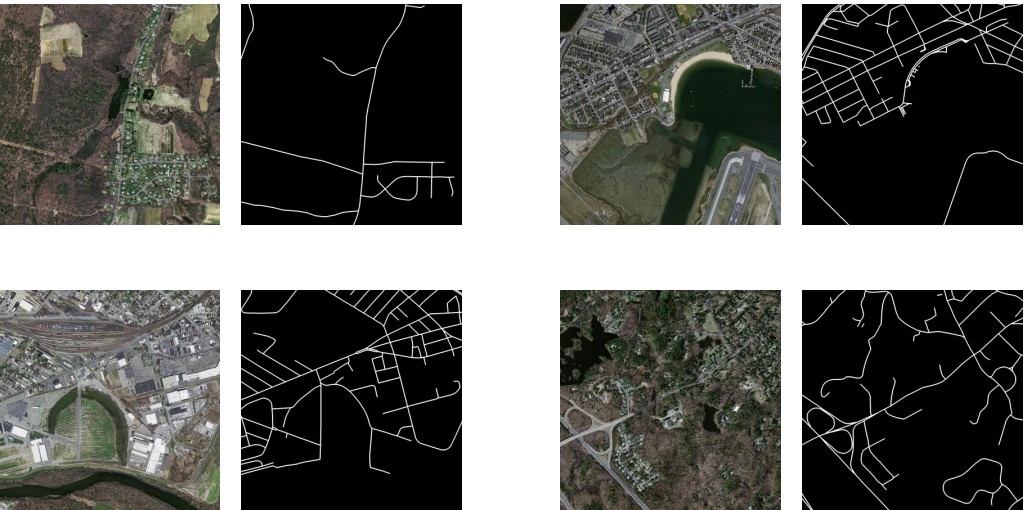

**Figure 2** **Overview of the data.** All images and masks from the Massachusetts dataset.

channels. Similarly, the original data were cropped into nonoverlapping images with a resolution of 512 × 512 pixels. Finally, 3,744 images were used for training, 30 images were used for validation, and 196 images were used for testing. Moreover, all images with large white blank areas were removed manually.

## Details of the model structure

We propose Dual Path Information Fusion Network (DPIF-Net) to improve the performance of rural road extraction by exploring the potential of combining the capabilities of transformers and CNNs for road segmentation. The schematic structure of DPIF-Net is displayed in Fig. 3. First, the top encoder branch uses a transformer to model global road information in the input remote sensing image, while the other encoder branch uses convolution operations to extract local details of roads and process spatial and channel information. Second, the feature information of the two branches is effectively fused. Finally, each layer of the decoder fuses high-level features from the previous layer with low-level features from the convolutional branch and gradually upsamples the image to the original resolution to obtain a binary image containing only roads.

## Local detail information encoder based on a CNN

In DPIF-Net, we propose a convolution module called the local detail feature extraction (LDFE) block as shown in Fig. 4. This block is composed of three parts to efficiently extract road features while keeping the number of network parameters low.

In part A, a traditional 3 × 3 convolution is applied to the input feature map to extract preliminary feature information without altering its resolution, resulting in a feature map that is four times larger than the input in the channel dimension. Then, the obtained features are spliced and fused using a skip connection.

In part B, each channel of the feature map is processed separately using a 3 × 3 convolution to extract semantic feature information, and a leaky-ReLU (*Maas, Hannun &*
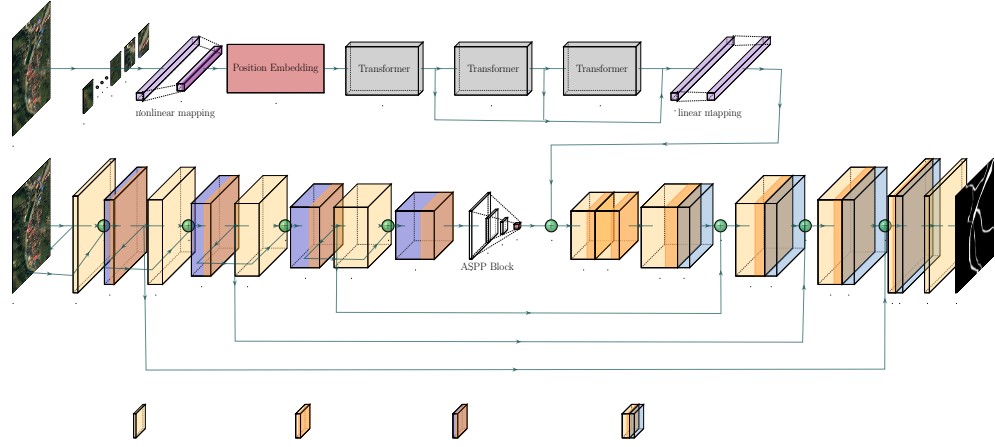

**Figure 3** **Overview of the proposed model for rural road extraction.** All satellite images and masks from the Massachusetts dataset.

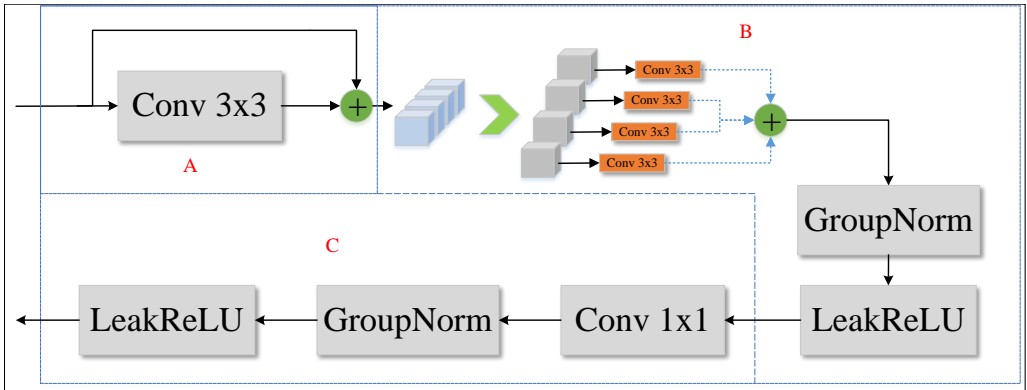

**Figure 4** **Local detail feature extraction block.**

*Ng, 2013*) activation layer is used to obtain nonlinear features, as expressed in Eq. (1).

$$Leak - ReLU(x) = \begin{cases} x & (x > 0) \\ \alpha x & (x <= 0) \end{cases} \tag{1}$$

$\alpha$ is a small gradient value, which is set to 0.2 in this article.

In part C, the feature information from part B is combined, cross-channel interaction is achieved using a $1 \times 1$ convolution, and then another leaky-ReLU activation layer is applied.

## Global information encoder based on a Transformer module

The ASPP module was originally introduced to increase the receptive field without sacrificing too much resolution, allowing for the preservation of image details as much as possible (*Chen et al., 2018a*). To process multiscale road feature information extraction,

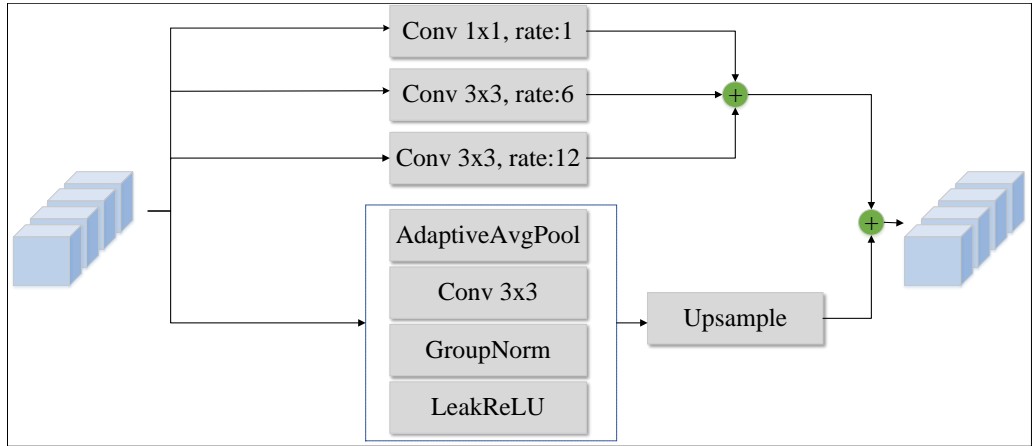

**Figure 5** **Modified ASPP module.**

we propose improvements to the ASPP module. Specifically, there are three different dilated convolution modules and a global adaptive pooling module, which can extract more feature information, as shown in Fig. 5. Moreover, different dilation ratios are used to obtain abstract feature information at different scales, which provides a basis for the subsequent integration of features and accurate road segmentation. In this way, the modified ASPP module obtains more abstract semantic information and road topology information and achieves better robustness; it can accept input feature images of any size and finally produce output of a fixed size.

In addition to the improvements to the CNN module proposed above, a Trans block is proposed to employ the Transformer architecture at the beginning of the model to capture global context information while avoiding interference with the CNN branch. Specifically, the Trans block can compensate for the shortcomings of the CNN in capturing global context information.

To process image blocks with a transformer, we convert them into one-dimensional data by using a fully connected (FC) layer. Due to the expansion into one-dimensional data, the position information is lost; therefore, the input feature data are converted from index numbers into a one-hot encoding matrix to maintain the position relationships of the sequence. Moreover, a random weight matrix is right-multiplied to complete the input position embedding. This method ensures that each token in the sequence has a unique position representation, which is crucial for the transformer to capture long-range dependencies.

The schematic structure of an individual transformer block is shown in Fig. 6 (*Dosovitskiy et al., 2020*). The Transformer encoder mainly consists of alternating multihead self-attention (MSA) layers and multilayer perceptron (MLP) blocks. Before these two modules, layer normalization (LN) is applied for normalization, and a residual connection structure is used. The MLP block consists of two FC layers, in which the Gaussian error linear unit (GeLU) function is applied for nonlinear activation to obtain nonlinear features. The definition of the GeLU (*Hendrycks & Gimpel, 2016*) activation function is shown in Eq. (2).

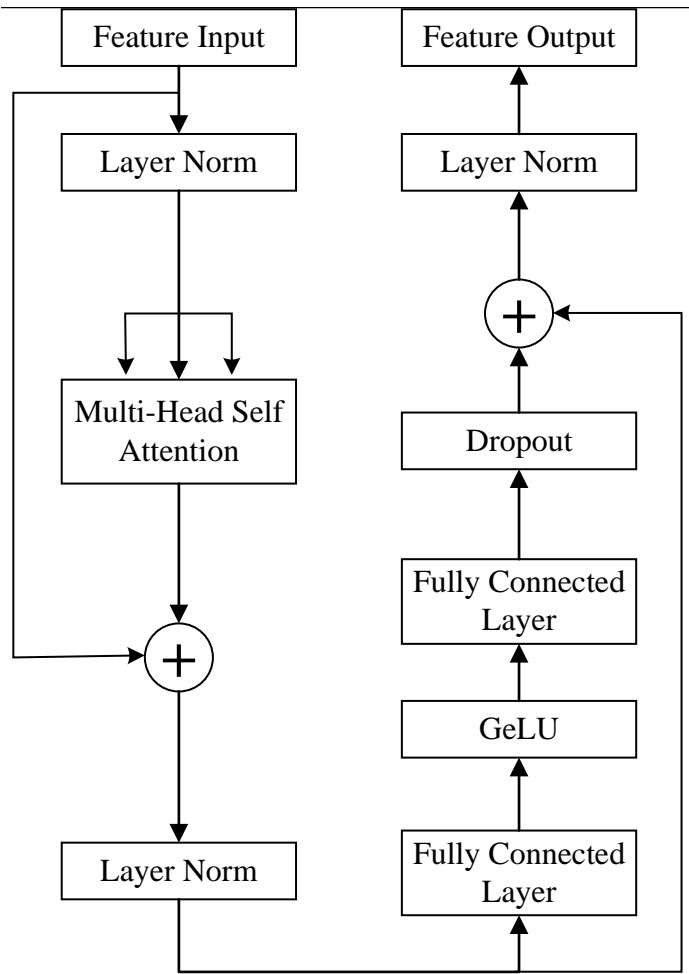

**Figure 6** **The structure of the transformer block.**

$$GeLU(x) = xP(X \leq x) = x \times \phi(x) X \sim N(0,1) \tag{2}$$

$x$ is the input, and X is a random variable that follows a Gaussian distribution with mean 0 and variance 1. $P(X \leq x)\phi(x)$ is the cumulative distribution of the Gaussian normal distribution of $x$, for which there is no analytical expression; instead, its approximate calculation method is shown in Eq. (3).

$$GeLU(x) = \frac{1}{2}x \left(1 + \tanh\left(\sqrt{\frac{2}{\pi}}(x + 0.044715x^3)\right)\right). \tag{3}$$

In our work, the number of attention heads is set to 16, assuming that the input image is $x$ ($x \in i^{H \times W \times C}$), where H, W and C are the height, width and channels, respectively, of $x$. The original image block feature $x_p$ ($x_p \in i^{N \times P^2 \times C}$) is obtained by expansion into one-dimensional data, $x_p \in i^{N \times P^2 \times C}$, where N is the number of small P ×P patches into

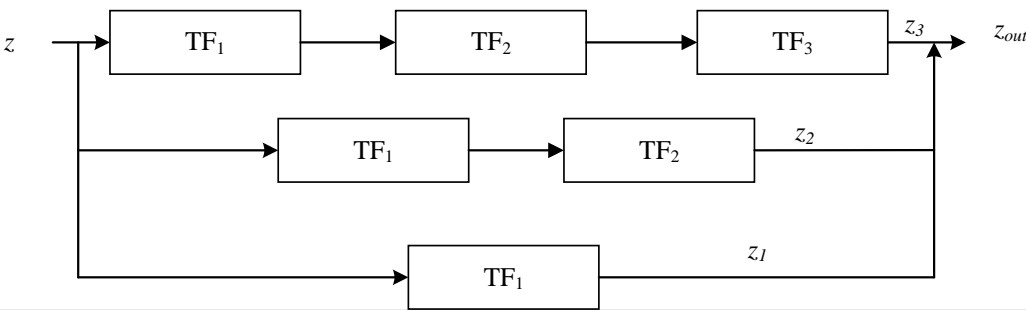

**Figure 7  Trans block structure.**

which the original input image is cropped. N and P satisfy the relationship shown in Eq. (4).

$$N = \frac{H \times W}{P^2}. \tag{4}$$

Then, after dimension embedding and nonlinear mapping of the GeLU function, the i-th image block feature $z^i$ can be expressed as shown in Eq. (5).

$$z^i = GeLU(x_p^i E) \tag{5}$$

$E \in i^{P^2 \times C \times D}$. After the corresponding position encoding, the vector $z$ input into the transformer (TF) can be expressed as shown in Eq. (6).

$$z = \left[ GeLU(x_p^1 E), GeLU(x_p^2 E), \ldots, GeLU(x_p^N E) \right] + E_{pos} \tag{6}$$

$E_{pos} \in i^{N \times D}$. Finally, the input vector $z$ is subjected to the calculations shown in Eq. (7) and Eq. (8).

$$z' = MSA(LN(z)) + z \tag{7}$$

$$z_1 = MLP(LN(z')) + z'. \tag{8}$$

Multiple TF blocks are used to build the Trans block, as shown in Fig. 7, and the generated output of global road information from the original remote sensing image, $z_{out}$, is expressed as shown in Eq. (9).

$$z_{out} = z_1 + z_2 + z_3 \tag{9}$$

## Decoder based on context information fusion

After the two different kinds of information discussed above are obtained, the final branch combines the two different kinds of road semantic information to fuse the global and local road features, as shown in part A of Fig. 8. The high-level feature information is extracted using two $3 \times 3$ convolutional layers without changing the feature map resolution. Group normalization (GN) is used to normalize the features, and Leaky-ReLU is used as the
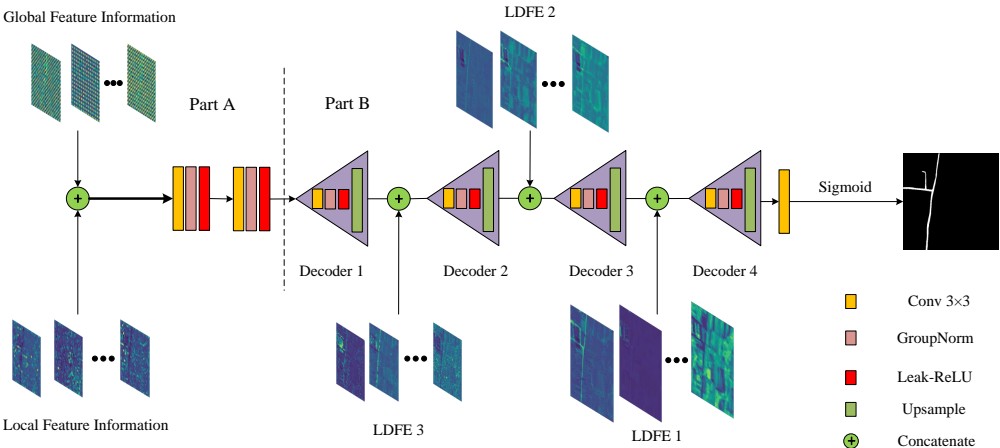

**Figure 8** **Information fusion strategy.**

nonlinear activation function to capture the nonlinear characteristics of the roads. This process fully integrates the semantic information from the two different sources, resulting in more comprehensive road semantic feature information that covers the feature information of various road categories.

To solve the problem that the spatial location information of the roads is not obvious due to multiple convolutions, this article adopts the design concept of the U-Net network structure and employs skip connections in the decoder to transfer the low-level feature information from the output of the LDFE module to the corresponding decoder port for splicing, as depicted in part B of Fig. 8. Subsequently, the features are fused layer by layer *via* 3 × 3 convolutional layers, and the feature map's resolution is incrementally restored *via* bilinear interpolation until it finally reaches the original image resolution.

Finally, *Wu & He (2018)* proposed group normalization (GN) to improve the efficiency of training rather than batch normalization (BN) BN works effectively for a relatively large batch size. However, a small batch size leads to inaccurate estimation of the batch statistics, and reducing the batch size for BN dramatically increases the model error. GN can achieve approximately the same accuracy performance as BN for a moderate batch size and outperforms other normalization variants because it can still achieve a small error rate even when the batch size undergoes large fluctuations. The GN calculation is expressed as shown in Eq. (10): (*Wu & He, 2018*).

$$y_i = \frac{\gamma}{\sigma_i}(x_i - \mu_i) + \beta \tag{10}$$

$\gamma$ and $\beta$ are trainable scale and shift parameters, respectively, and $\mu_i$ and $\sigma_i$ in Eq. (10) are the mean and standard deviation computed as shown in Eq. (11): (*Wu & He, 2018*)

$$\mu_i = \frac{1}{m}\sum_{k \in S_i} x_k, \sigma_i = \sqrt{\frac{1}{m}\sum_{k \in S_i}(x_k - \mu_i)^2 + \varepsilon} \tag{11}$$

$\varepsilon$ is a small nonzero constant. $S_i$ is the set of pixels, and $m$ is the size of $S_i$.

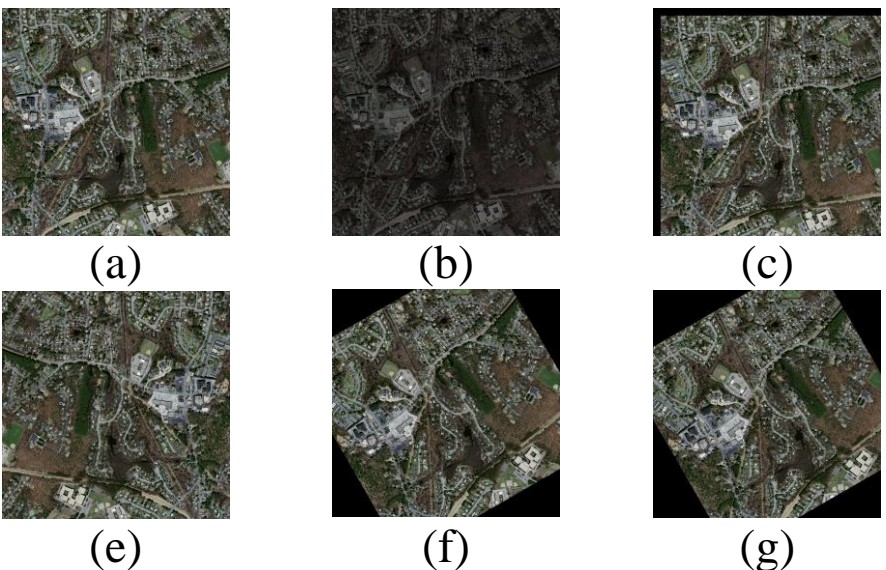

**Figure 9  Data augmentation strategy.** (A) Original image, (B) HSV color jitter, (C) translation, (D) flip, (E) random rotation, (F) translation and rotation. (All satellite images and masks from the Massachusetts dataset).

## Experimental design description

DPIF-Net was trained on an RTX 3090 GPU. At the beginning of training, we utilized common data augmentation techniques, including translation, rotation, flipping, scaling, and random color jitter as shown in Fig. 9, which contributed to improving the model's robustness and performance. The model implementation was based on PyTorch, the optimizer for all structures was Adam, the initial learning rate was set to 0.0002, and the batch size during training was set to 2. The MSELoss was used to calculate the loss.

To assess the performance of DPIF-Net in rural road extraction, we conducted three primary comparative experiments across distinct datasets. Each of these comparative experiments entailed a juxtaposition between mainstream models and ours. In addition, the performance of DPIF-Net was compared with that of the U-Net, SegNet, D-LinkNet, and DeepLabv3+ models on the above three road datasets. In these experiments, our goal was to observe its capabilities for extracting roads in complex scenarios through evaluation metrics, assessing both the completeness and accuracy of road extraction. In the discussion, we assessed the parameter sizes among the models and the visual assessments of both completeness and accuracy. Furthermore, the ablation experiments are conducted to observe the contributions of its two branches, which are designed to identify which modules are most crucial for DPIF-Net's performance.

## Experimental evaluation metrics

To evaluate the effectiveness of our model, four common metrics are selected: intersection over union (IoU), precision, recall, and $F_1$ score ($F_1$). High IoU indicates the model's accuracy in predicting the location and shape of roads, which is particularly crucial for

assessing the model's ability to recognize roads (*Lian et al., 2020*). The IoU is calculated as shown in Eq. (12).

$$IoU = \frac{TP}{TP + FP + FN} \tag{12}$$

TP, FP, TN, and FN denote true positives, false positives, true negatives, and false negatives, respectively.

In the context of rural roads, where non-road areas often constitute a significant proportion, leading to an imbalance between positive and negative samples in the dataset, $F_1$ score becomes particularly crucial for assessing performance under such conditions, which are calculated as shown in Eq. (13).

$$F_1 = \frac{2 \times precision \times recall}{precision + recall} \tag{13}$$

The precision and recall are employed to evaluate the model's capability to correctly identify roads, which are calculated as shown in Eq. (14) and Eq. (15), respectively.

$$precision = \frac{TP}{TP + FP} \tag{14}$$

$$recall = \frac{TP}{TP + FN}. \tag{15}$$

## RESULTS

### Road extraction experiment based on our road dataset

The results on our dataset are shown in Fig. 10. Through detailed comparisons, we find that in Examples 1–2 and Examples 4–7, DPIF-Net generates fewer broken road segments (misidentified connected vectors) than the other four models. At the same time, the occluded parts caused by trees can be completely extracted to ensure the connectivity of the roads. In Example 3, DPIF-Net extracts the least erroneous information, yielding results almost consistent with the ground truth, whereas the other four models incorrectly extract road information in this example.

More detailed comparison results are presented in Table 1. DPIF-Net achieves the best performance in two metrics, namely, IoU and $F_1$ score, reaching 61.40% and 76.08%, respectively. In addition, DPIF-Net reaches 77.27% precision, which is 1.05% lower than the highest score and 0.6% lower than U-Net's score. DPIF-Net also reaches 74.94% recall, which is 0.08% lower than D-LinkNet's highest recall score of 75.02%, representing a small, almost negligible fluctuation.

Compared with U-Net, the IoU, recall and $F_1$ values of DPIF-Net are increased by 3.34%, 5.41%, and 2.61%, respectively. Compared with DeepLabv3+, the IoU, recall and F1 values are increased by 4.8%, 7.83% and 3.8%, respectively. Compared with D-LinkNet, the IoU, precision and F1 values increased by 0.35%, 0.64%, and 0.26%, respectively. Compared with SegNet, the IoU, precision, recall and F1 values are increased by 7.52%, 1.52%, 9.83%,

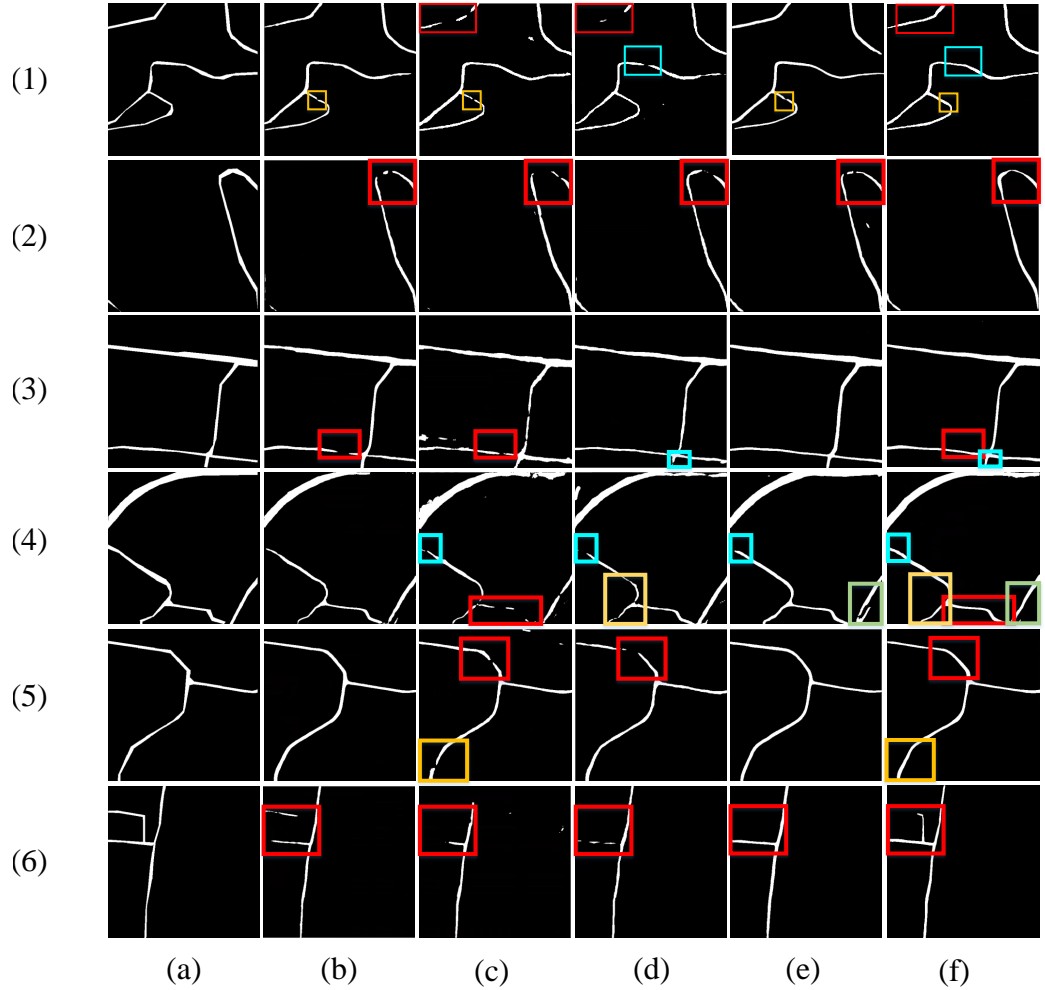

(1) (2) (3) (4) (5) (6)

(a) (b) (c) (d) (e) (f)

**Figure 10** **Road extraction results for our dataset.** (A) Satellite images. (B) Real masks. (C) U-Net. (D) SegNet. (E) DeepLabv3+. (F) D-LinkNet. (G) Ours.

**Table 1** **Comparison of different road extraction methods on the GF2 road dataset.**

|  | IoU | Precision | Recall | $F_1$ |
|---|---|---|---|---|
| U-net | 0.5806 | 0.7788 | 0.6953 | 0.7347 |
| Deeplabv3+ | 0.5660 | **0.7832** | 0.6711 | 0.7228 |
| D-Linknet | 0.6105 | 0.7663 | **0.7502** | 0.7582 |
| Segnet | 0.5388 | 0.7575 | 0.6511 | 0.7003 |
| Ours | **0.6140** | 0.7727 | 0.7494 | **0.7608** |

**Notes.**
The best results are highlighted in boldface.

and 6.05%, respectively. The largest increases in IoU, precision, recall, and F1 score reach 7.52%, 1.52%, 9.83%, and 6.05%, respectively. In the extraction of rural roads, the crucial aspects lie in correctly identifying roads and preserving their completeness, both reflected in

**Table 2   Comparison of different road extraction methods on DeepGlobe Roads Dataset.**

|  | IoU | Precision | Recall | $F_1$ |
|---|---|---|---|---|
| U-net | 0.5233 | 0.7468 | 0.6362 | 0.6870 |
| D-Linknet | 0.5407 | 0.5767 | **0.8962** | 0.7018 |
| Deeplabv3+ | 0.3692 | 0.5428 | 0.5359 | 0.5393 |
| Segnet | 0.5640 | 0.6559 | 0.8009 | 0.7212 |
| Ours | **0.5700** | **0.7476** | 0.7058 | **0.7261** |

**Notes.**
The best results are highlighted in boldface.

the IoU and F1 metrics. DPIF-Net achieves the highest IoU and F1 on the lower level roads, highlighting its superiority in extracting rural roads, which underscores its proficiency in correctly extracting rural roads and recognizing road shapes.

## Road extraction experiment based on the DeepGlobe road dataset

A second experiment was conducted on the DeepGlobe road dataset, and the results of the visual assessment comparison are presented in Fig. 11. Figure 11 shows the detailed effects of rural road segment extraction for 8 examples. In the first example, compared with U-Net and SegNet, DPIF-Net extracts more complete road information. In contrast, D-LinkNet produces more mistakenly extracted road segments and more fractures than DPIF-Net. In the 2nd to 6th examples, the extraction results of U-Net and SegNet show more broken segments, while DPIF-Net achieves almost the same road integrity as D-LinkNet, whereas D-LinkNet misextracts more road segments than DPIF-Net. In the 7th and 8th examples, the integrity of the roads extracted by DPIF-Net is higher than that of the other three methods.

The detailed evaluation index comparisons are shown in Table 2. From the indicator data in Table 2, it can be seen that the model with the worst comprehensive performance is DeepLabv3+, which has the lowest score in all metrics. Its recall is lower by more than 30%, and the other three metrics are lower by nearly 20%. DPIF-Net achieves the best results in terms of IoU, precision and $F_1$ score, reaching values of 57%, 74.76% and 72.61%, respectively. However, its recall is lower than those of D-LinkNet and SegNet at only 70.58%. D-LinkNet has the highest recall score of 89.62%.

In summary, DPIF-Net improves the IoU by 0.6–20.08%, the precision by 0.08–20.48%, and the $F_1$ score by 0.49–18.68% on the DeepGlobe road dataset. Its recall is weaker than those of SegNet and D-LinkNet but 6.96% higher than that of U-Net and 16.99% higher than that of DeepLabv3+.

## Road extraction experiment based on the Massachusetts road dataset

To further test the generalization ability of the proposed DPIF-Net, a similar experiment was carried out on the Massachusetts road dataset. The visual assessment of the compared methods on this dataset is shown in Fig. 12. All the models almost completely extract the road information, but from Fig. 12, it can be seen that the other models do not extract some road details completely enough, resulting in various fractures. Comprehensive comparisons show that the proposed model is better than the others in extracting many details.

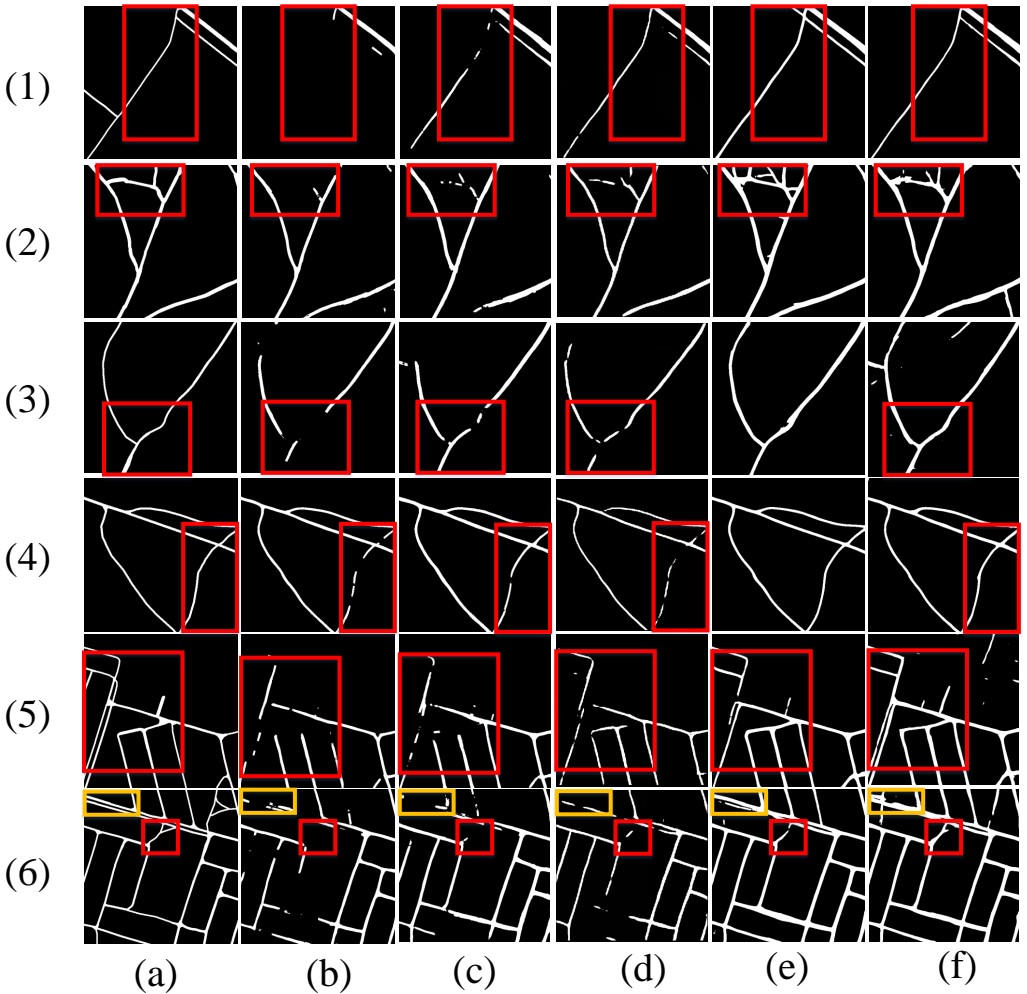

**Figure 11** **Road extraction results for the DeepGlobe road dataset.** (A) Ground truth. (B) U-Net. (C) SegNet. (D) DeepLabv3+. (E) D-LinkNet. (F) Ours.

Table 3 displays the detailed road extraction results on the Massachusetts road dataset, revealing that DPIF-Net surpasses the other models in terms of IoU, precision, and F1 score, with values of 53.82%, 82.48%, and 70%, respectively. Although DeepLabv3+ achieves the highest recall score of 63.92%, the IoU, F1 score and precision of DeepLabv3+ are significantly lower than those of all other models, with differences of 10–40%.

Compared to other models, DPIF-Net integrates global and local information more extensively, enabling it to capture more features and thereby enhance its recognition capabilities. DPIF-Net achieves superior performance in predicting road accuracy and completeness, demonstrated by attaining maximum values in IoU, precision, and F1 on the DeepGlobe and Massachusetts datasets, showcasing its advantages in overall road extraction.

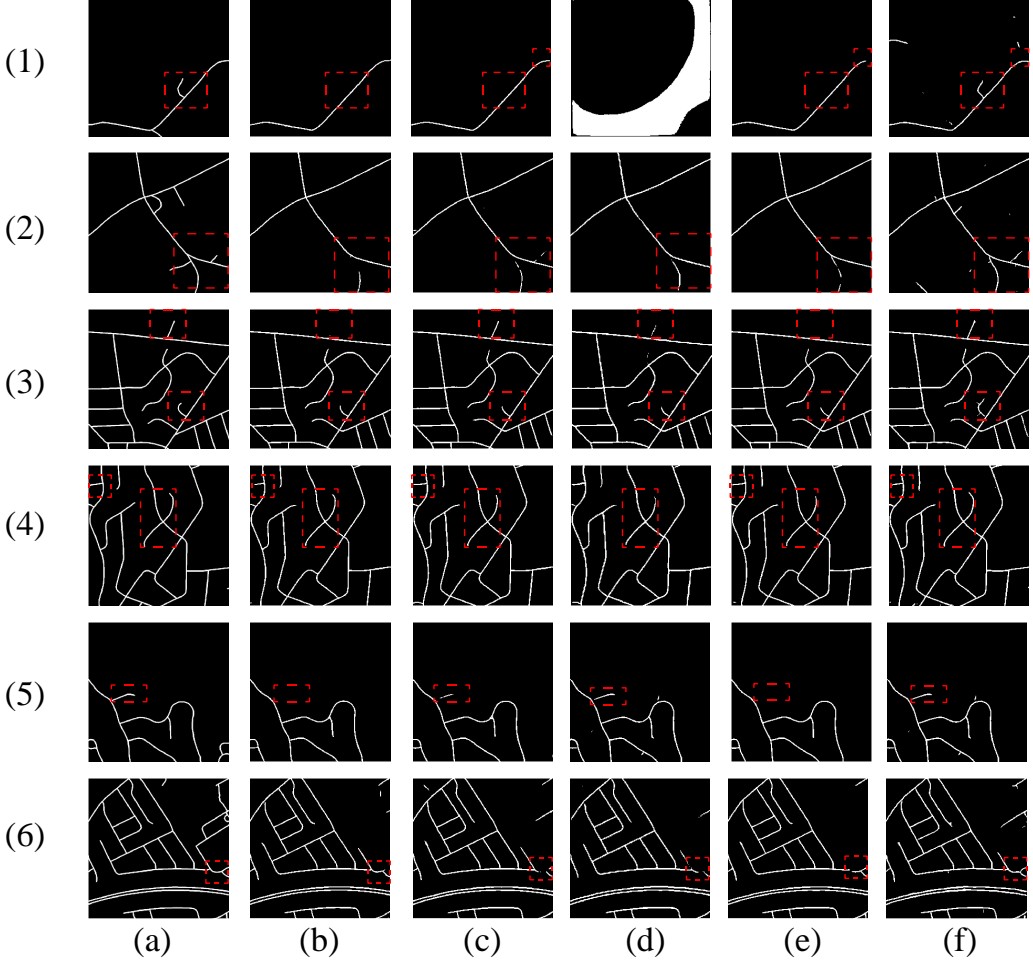

**Figure 12** **Road extraction results on the Massachusetts road dataset.** (A) Ground truth. (B) U-Net. (C) SegNet. (D) DeepLabv3+. (E) D-LinkNet. (F) Ours.

**Table 3** **Comparison of different road extraction methods on Massachusetts Roads Dataset.**

|  | IoU | Precision | Recall | $F_1$ |
|---|---|---|---|---|
| U-net | 0.4265 | 0.5698 | 0.6290 | 0.5980 |
| Deeplabv3+ | 0.3115 | 0.3779 | **0.6392** | 0.4750 |
| D-Linknet | 0.4260 | 0.5668 | 0.6316 | 0.5975 |
| Segnet | 0.4197 | 0.5632 | 0.6223 | 0.5913 |
| Ours | **0.5382** | **0.8248** | 0.6077 | **0.7000** |

**Notes.**
The best results are highlighted in boldface.

## DISCUSSION

In this section, we provide a detailed discussion on several important aspects of DPIF-Net. First, we elaborate on the input and output mechanisms within the network, highlighting the various components and their respective roles. Second, we discuss the significance of

the network architecture for road extraction and how DPIF-Net utilizes the capabilities of both CNNs and transformers for effective feature representation and information fusion. Moreover, we present a comprehensive parameter comparison of DPIF-Net with other state-of-the-art models for road extraction, demonstrating the effectiveness and efficiency of our proposed model. Finally, we report an ablation study conducted to analyze the role of each branch of DPIF-Net, which provides insights into the contribution of each component toward the final performance of the model.

## Evaluation of the generalization performance and road representation ability of the proposed model

First, we evaluate the generalization performance and road representation ability of the proposed model. To gain deeper insight into the inner workings of DPIF-Net, feature heatmaps for five different stages in the decoder are displayed in Fig. 13 to illustrate how the model extracts roads. The results from the three different datasets indicate that DPIF-Net effectively learns road features with clear boundaries. The feature maps show that the encoder learns various levels of feature representations of the input image, and the decoder combines these representations to generate accurate road masks. As the decoder performs upsampling four times, the semantic information of the extracted roads becomes increasingly abstract, which highlights the ability of DPIF-Net to learn high-level features. These results demonstrate that DPIF-Net not only has excellent road extraction performance but also possesses good robustness and feature representation capabilities; thus, it shows promising potential for various applications in remote sensing image analysis.

## Discussion on the quantity of model parameters

Furthermore, a comparison of the parameters used in the experiments for each model provides insight into their respective strengths and weaknesses. Table 4 presents a comprehensive overview of the parameters for each model, indicating that DPIF-Net has the fewest parameters, with a data volume of only 63.9 MB, which is similar to that of U-Net. While D-LinkNet achieves better feature extraction in some cases, it also has a significantly larger number of parameters due to its use of ResNet101 as the encoder and deeper network layers. On the other hand, DeepLabv3+ also has a high number of parameters due to the use of Xception as the encoding network, but it performs poorly in road extraction experiments. SegNet, with approximately twice as many parameters as DPIF-Net, also yields inferior experimental results for road segmentation. These comparisons highlight the trade-off between the number of parameters and the effectiveness of a model for road extraction tasks. While having more parameters may improve feature extraction, it also increases a model's complexity and computational cost. DPIF-Net, with its simple yet effective structure and relatively few parameters, proves to be a promising model for road extraction from remote sensing images.

## Ablation study

To better understand the contribution of each branch in the encoder part of DPIF-Net to the road extraction results, an ablation study was conducted using our dataset. A comparison

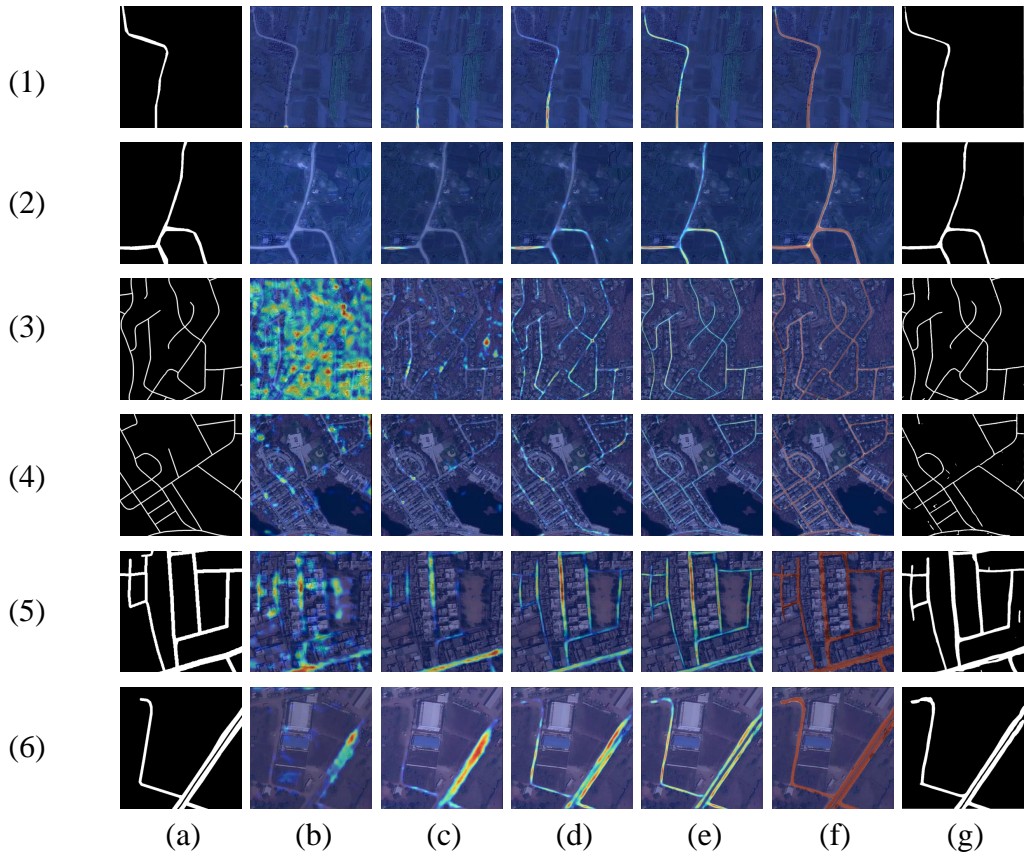

**Figure 13** **Visualization of features at different levels.** (A) Ground truth. (B) First decoder output. (C) Second decoder output. (D) Third decoder output. (E) Fourth decoder output. (F) Last convolution output. (G) Final extraction result. The data sources are (1)–(2) our dataset, (3)–(4) the Massachusetts road dataset, and (5)–(6) the DeepGlobe road dataset.

**Table 4** **Comparison of parameters of each model.**

| Model name | Parameter quantity |
| --- | --- |
| U-net | 69.2MB |
| D-Linknet(resnet101) | 947.3MB |
| Segnet | 117.9MB |
| Deeplabv3+(xception) | 219.0MB |
| Ours | **63.9MB** |

**Notes.**
The best results are highlighted in boldface.

of the results reveals that the two branches make different levels of contributions to the road extraction results, as presented in Table 5. Specifically, the branch that incorporates the CNN-based feature extractor makes a stronger contribution to the final road extraction results than the branch that employs the transformer-based feature extractor.

**Table 5  Comparison of contribution of encoder branches to road extraction.**

| CNN branch | Transformerbranch | IoU | precision | recall | F$_1$ |
|---|---|---|---|---|---|
| ✓ | ✗ | 0.5693 | **0.7871** | 0.6730 | 0.7256 |
| ✗ | ✓ | 0.4411 | 0.6764 | 0.5591 | 0.6122 |
| ✓ | ✓ | **0.6140** | 0.7727 | **0.7494** | **0.7608** |

**Notes.**
The best results are highlighted in boldface.

Notably, due to the strict parameter limitations of the final model presented in this article, only three transformer blocks were used in the transformer-based feature extractor. This resulted in suboptimal performance in comparison to the CNN-based feature extractor. However, the performance of the transformer-based feature extractor could be improved by stacking more transformer blocks, albeit at the cost of dramatically increasing the number of parameters.

While DPIF-Net has demonstrated impressive performance by effectively combining a CNN and a transformer, there are several potential areas for improvement in future research on road extraction using DPIF-Net. First, more advanced architectures for the transformer block could be explored to further enhance the model's performance without significantly increasing the number of parameters. Second, modified attention mechanisms or other forms of spatial information modeling might improve the model's ability to capture fine details and connectivity information in the road network. Third, methods to improve the accuracy of road boundary delineation and reduce false positive rates could further increase the model's utility in practical applications. Overall, these potential areas for improvement could help advance the state of the art in road extraction using deep learning models.

## CONCLUSIONS

In this study, we successfully constructed a dedicated lower level category roads and developed DPIF-Net. By effectively harnessing the strengths of both transformers and CNNs, our model has demonstrated excellent performance in road extraction tasks with in comparison with advanced models of the same period. It not only enhances extraction accuracy but also achieves higher levels of road connectivity. This achievement holds significant implications not only for the field of road extraction but also for increasing attention to rural road issues in research and government decision-making. Despite constraints on research time and workload, DPIF-Net can leverage the rapid advancements in deep learning to enhance its two branches by incorporating state-of-the-art transformer or CNN modules, thus further improving the model's performance. Additionally, examining the model's generalization capabilities across different geographical regions and complex weather conditions to validate its practicality is crucial. These efforts will contribute to the advancement of road extraction technology in remote sensing images, providing strong support for rural development and infrastructure construction.

## ACKNOWLEDGEMENTS

The authors would like to express their gratitude to the following individuals and organizations for their contributions to this research: DeepGlobe for making available a free road dataset and Volodymyr Mnih for providing a free road dataset.

### Funding

This work was supported by the Major Project of High Resolution Earth Observation System (Grant No."30-Y60B01-9003-22/23") and the China Scholarship Council (No. 202306070036). The funders had no role in study design, data collection and analysis, decision to publish, or preparation of the manuscript.

### Grant Disclosures

The following grant information was disclosed by the authors:
Major Project of High Resolution Earth Observation System: 30-Y60B01-9003-22/23.
China Scholarship Council: 202306070036.

### Competing Interests

The authors declare there are no competing interests.

### Author Contributions

- Yuan Sun conceived and designed the experiments, performed the experiments, analyzed the data, performed the computation work, authored or reviewed drafts of the article, and approved the final draft.
- Xingfa Gu conceived and designed the experiments, performed the experiments, analyzed the data, performed the computation work, authored or reviewed drafts of the article, and approved the final draft.
- Xiang Zhou conceived and designed the experiments, performed the experiments, analyzed the data, performed the computation work, authored or reviewed drafts of the article, and approved the final draft.
- Jian Yang conceived and designed the experiments, performed the experiments, analyzed the data, performed the computation work, authored or reviewed drafts of the article, and approved the final draft.
- Wangyao Shen conceived and designed the experiments, performed the experiments, analyzed the data, performed the computation work, prepared figures and/or tables, authored or reviewed drafts of the article, and approved the final draft.
- Yuanlei Cheng conceived and designed the experiments, performed the experiments, analyzed the data, performed the computation work, prepared figures and/or tables, authored or reviewed drafts of the article, and approved the final draft.
- Jin Ming Zhang analyzed the data, performed the computation work, prepared figures and/or tables, authored or reviewed drafts of the article, and approved the final draft.

- Yunping Chen conceived and designed the experiments, performed the experiments, analyzed the data, performed the computation work, authored or reviewed drafts of the article, and approved the final draft.

## Data Availability

The lowgrade dataset is available at Figshare:

jimmy, Zhang (2023). GF_LowGradeRoadDataset.zip. figshare. Dataset. https://doi.org/10.6084/m9.figshare.24457369.v2.

The DeepGlobe Road Extraction Challenge dataset is available at https://competitions.codalab.org/competitions/18467#learn_the_details-overview.

The Massachusetts road and building dataset is available at https://www.cs.toronto.edu/~vmnih/data/.

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
