# Peer review of "DPIF-Net: a dual path network for rural road extraction based on the fusion of global and local information"

_PeerJ Computer Science, doi:10.7717/peerj-cs.2079_

## Round 0.1 · original submission · Major Revisions

In this study, the authors present DPIF-Net, a convolutional neural network (CNN)-based approach for extracting roads. The study may help government agencies in charge of highways better comprehend their structure and make improvements to the road system.

The paper has a good contribution however a number of comments must be addressed by the author to make the paper in very good standard.

Reviewer 1 ·

Basic reporting

The article discusses a study conducted by researchers from the Aerospace Information Research Institute, Chinese Academy of Sciences, and the School of Automation Engineering, University of Electronic Science and Technology of China. The study aims to address the issue of automatic extraction of roads from remote sensing images, particularly focusing on low-grade roads in rural areas this study.

Experimental design

The study's experimental design is not explicitly detailed in the provided excerpts. However, it is mentioned that the researchers carried out comparative experiments on the DeepGlobe and GF-2 low-grade road datasets to test the performance and generalization ability of their proposed model, the Dual Path Information Fusion Network (DPIF-Net).

The article lacks a detailed description of the experimental design, including the specific steps taken in the experiments, the metrics used to evaluate the model's performance, and the statistical methods used to analyze the results. Providing this information would make the study more transparent and reproducible.

Validity of the findings

The article has a deficiency in providing an elaborate account of the experimental design, encompassing the precise procedures employed in conducting the tests, the metrics employed to assess the performance of the model, and the statistical techniques employed to analyze the obtained results.

Additional comments

no comments. I attched file regarding this artical.

Annotated reviews are not available for download in order to protect the identity of reviewers who chose to remain anonymous.

Reviewer 2 ·

Basic reporting

In this paper, the authors present a road extraction method DPIF-Net based on convolutional neural networks (CNNs). The research could be useful for the government authorities related to highways for better understanding the structure as well as road infrastructure improvements.
Although the abstract discusses the topic of extracting roads from remote sensing photos, it does not go into detail regarding the worldwide significance of this challenge or the potential real-world applications that could be aided by better road extraction methods
The creation of the GaoFen-2 (GF-2) low-grade road dataset is mentioned in the abstract. However, more information about the dataset, such as its size, diversity, and how it solves the difficulties of low-grade country roads, would be beneficial.
The proposed Dual Path Information Fusion Network's use of Transformers and convolutional neural networks (CNNs) is briefly mentioned in the abstract (DPIF-Net). While this provides a broad overview, readers interested in the technical details would benefit from more detailed

Experimental design

The study lacks reporting and comparison with the latest available extraction models.
Please also discuss the data set in the abstract
The creation of data set seems to be missing in the methods section
you have used DeepGlobe (Demir et al. 2018) and Massachusetts (Mnih 2013) datasets for experimentation, but don't you feel that they are too old datasets as the country is rapidly changing and adding many new additions.

Validity of the findings

The major missing part of the results/findings is the validity of the proposed method, as I cannot see any justification for how the method has been validated for its claims.
The abstract makes a strong argument about having the highest IoU and F1 score compared to other approaches. To quantify the performance gain, though, particular numerical values or percentages would be more illuminating.

Additional comments

There are lots of language and grammatical issues , therefore, the authors are advised to address them carefully and give a full professional proof read, some of the issues are being highlighted here

in abstract F1 score on three datasets should be F1 scores on three datasets
In abstract we construct the GaoFen-2 (GF-2) should be we constructed the GaoFen-2 (GF-2)
On line 96 Their method achieved good performance on medical image segmentation could be Their method achieved good performance in medical image segmentation

2. The figures are not of good quality eg. Figure 2 Overview of the proposed model for rural road extraction is very blur and hard to understand and interpret.
please also justify the overall benefit of the proposed research ?

---

## Round 0.2 · Minor Revisions

The difficulties encountered in creating the dataset are not mentioned; this is typical behaviour for undertakings of this nature.

More information is left out, and the contribution to the model's overall performance has to be explained.

Give more details about the metrics you have applied to real-world DPIF-Net settings.

The authors have addressed the majority of the reviewers' criticisms from the earlier rounds, but minor remarks still need to be addressed.

Reviewer 1 ·

Basic reporting

NO comment

Experimental design

No comment

Validity of the findings

No comment

Additional comments

The author has responded to all the questions, and based on this, I accept the work

Reviewer 2 ·

Basic reporting

Provide additional details on how the GaoFen-2 rural road dataset was constructed. Include information on the image sources, data preprocessing steps, and any challenges faced during the dataset creation

Experimental design

Offer more insights into the validation experiments. What specific aspects of low-grade road extraction do these experiments validate, and how do they contribute to the model's overall performance?

Validity of the findings

While you mention comparative experiments with U-Net, SegNet, DeepLabv3+, and D-LinkNet, briefly highlight the key differences in performance or methodology that make DPIF-Net stand out
When presenting results, explain the significance of the achieved IoU and F1 scores. How do these metrics translate into practical benefits for rural road extraction?

Additional comments

Mostly the comments are addressed but needs some minor changes

---

## Round 0.3 · Minor Revisions

As you see I have taken over the editing of your manuscript, and after an extensive read, I found the manuscript and figures very interesting - especially as I think that also in my current work country (in Chile) lots of minor roads may still be unmapped. Although the OpenStreetMap community has done a very good job in manual mapping (could be another interesting database to compare completeness). Anyway from what I see there are only minor things left to improve. My detailed comments are as follows:

- Abstract:
-- I would advise to replace "millions of kilometers" with "thousands" (or hundreds) if you don't have some numbers on this. Millions sounds a bit unlikely to me, but...
-- "low grade datasets" >> minor roads in datasets or "low level category road datasets"

- Introduction:
-- 80: "Sections 5 and 6 are the final discussion and conclusion" >> Sections 5 and 6 provide a discussion and conclusions.

- Literature review:
--- 176: there is no closing statement/conclusions outlining a research gap or arguing, based on the presented review, why your work is needed. (I.e.you aren't showing how these cited works are affecting yours, so far). So please add this. Als revise the text for abbreviations that are not explained, such as RCNN, GAN, MLP : explain in a sentence or footnote what are transformer model is (I think I have not read an explanation later on)

- Materials & methods:
-- 184: "road conditions" >> "road categories"
-- 184: Missing literature reference to the GF-2 satellites sensors and/or products; also you says panchromatic, but also multispectral, so what wavelengths are we talking about here? (I though "panchromatic" is visible light spectrum, and multispectral is several bands... so please correct or explain - with the a link to a sensor/satellite reference)
-- 206: "High grade" >> "higher level"
-- Why you chose only 40 validation and 40 test samples if you have 5500 training samples? To me this sounds like preparing for overfitting (although you show later that it seems not overfit). How did you chose those numbers and also the numbers for the other 2 datasets (DeepGlobe + Massachusetts)? Also: How did you validate/did the error calculations? Did you do compare pixels or did you compare generated road-segment vectors with ground truth vectors? I ask this as in your case, both is possible.
-- what bands has/are the DeepGlobe data? Why do you use a 256pix overlap if the image has onla a size of 512px?
--- 239: not the "paper" but "you" (we) propose
--- 239: i don't see DPIF-Net mentioned before... so where does it come from etc? Any literature reference?
--- 249 where are the content/the blocks of figure 4 located in Figure 3?
--- 249 is there legend text missing in Figure 3? referring here to the coloured planes shown in the lower part of the picture?
--- What type of semantic features are extracted? Please add examples.
--- 260 What is "ReLu"? What means leaky-ReLu? (Footnote?)
--- 302 Please add sources/references from where you obtained the equations for GeLU (2) or is this your original work?
--- 311: this paper >> this work
--- 312: parameter C of the equation is not explained (I assume its colour?)
--- 359: The text that is written previously in this sentence, is this obtained from Wu & He? If so, please add the reference again
--- 359: Are the following equations (10+11) your work or from other authors? If the latter is the case, please add the reference.
---391/392: please add a reference/RS text book reference for the error metrics (to also indicate that they are indeed common in Remote Sensing).
--- 408: Figures 2,9 and 10: please try to add column headers directly to the figures (not just "a","b","c" etc.). Having the text separately elsewhere reduces readability and interpretability a lot (you could just add a new row and put the text inside, of this is a table)

- Results:
--- 419: real labels >> ground truth
--- 439: low grade road dataset >> "lower level roads" or "minor roads" (=> please revise this also in the abstracts)
--- thanks for the informative tables and graphics

- Discussion:
--- I like Figure 13!
--- interesting and worthwhile reading the discussion on the parameters - I didn't know the sets are that big (Table 4)
--- in table 4 you write DPIF-Net while in other tables you write "ours". Please be consistent and ideally replace "ours" by the algorithms name.

- Conclusions:
--- 571: low grade road datatset >> lower level category roads (or "minor roads")
--- 573: "extraction tasks with advanced models" >> "extraction tasks in comparison with advanced models"

Finally, how about a note of the availability of code or models? Are there any thoughts already?

Btw. another interesting future work would be if the algorithm could tag/identify what level/category the road may be - or how wide. As most road categories are defined based on traffic flow, width and surface type.

---

## Round 0.4 · accepted · Accept

Dear Authors, thank you for addressing all my comments and also for related improvements. I am happy with the manuscript in general. I saw however 2-3 spots were the English may be improved (see attached PDF).